# Effect of the Absence of α Carbonic Anhydrase 2 on the PSII Light-Harvesting Complex Size in *Arabidopsis thaliana*

**DOI:** 10.3390/plants14101529

**Published:** 2025-05-20

**Authors:** Elena M. Nadeeva, Natalia N. Rudenko, Lyudmila K. Ignatova, Daria V. Vetoshkina, Boris N. Ivanov

**Affiliations:** Institute of Basic Biological Problems, Federal Research Center, Pushchino Scientific Center for Biological Research of the Russian Academy of Sciences, 142290 Pushchino, Russia; nataliacherry413@gmail.com (N.N.R.); lkign@rambler.ru (L.K.I.); vetoshkinadv@gmail.com (D.V.V.); ivboni@rambler.ru (B.N.I.)

**Keywords:** carbonic anhydrase, *Arabidopsis thaliana*, photosynthesis, light-harvesting complex, hydrogen peroxide, thylakoid membrane

## Abstract

The absence of α-carbonic anhydrase 2 (α-CA2) in *Arabidopsis thaliana* leads to higher contents of chlorophylls *a* and *b*, and to a reduced chlorophyll *a*/*b* ratio, suggesting an increased PSII antenna compared to the wild type (WT). The evaluation of the OJIP kinetics of chlorophyll fluorescence in leaves of WT and α-carbonic anhydrase 2 knockout (α-CA2-KO) plants revealed higher apparent photosystem II (PSII) light-harvesting antenna size in the mutants. The higher levels of both Lhcb1 and Lhcb2 proteins in α-CA2-KO plants compared to WT plants were demonstrated using immunoblotting. Gene expression analysis showed increased *lhcb1* expression levels in mutants, whereas the *lhcb2* and *lhcb3* genes were downregulated. The content of hydrogen peroxide (H_2_O_2_) in leaves, as well as the production of H_2_O_2_ within the thylakoid membranes (“membrane” H_2_O_2_) was lower in α-CA2-KO plants as compared with WT plants. The expression levels of the genes encoding regulating proteins, which are involved in retrograde chloroplast–nucleus signaling, were lower in the α-CA2-KO than in the WT. The changes in the PSII light-harvesting complex size in the absence of α-CA2 correlates with the decreased accumulation of H_2_O_2_ in the leaves of mutants. It is suggested that this led to lower expression levels of the genes related to retrograde signal transduction from the chloroplast to the nucleus. The results of this study support previous conclusions regarding the involvement of α-CA2 in photosynthetic processes and its location within the chloroplasts of Arabidopsis.

## 1. Introduction

In the process of photosynthesis, CO_2_ molecules are initially incorporated into carbohydrate molecules and then into organic molecules as a result of biochemical transformations. Carbonic anhydrases (CAs) are able to participate both in the entry of CO_2_ molecules from the environment into photosynthesizing cells, and in the act, possibly supply them directly to the enzyme Rubisco, which provides ribulose bisphosphate carboxylation in the Calvin–Benson cycle. CAs catalyze the reversible reaction: CO_2_ + H_2_O ↔ HCO_3_^−^ + H^+^. They are classified into eight families: α, β, γ, δ, ζ, η, θ, and ι [1,2,3,4,5,6,7,8,9]. In plants, the representatives of three of them: α, β, and γ were detected [10,11]. Various CAs were found in different compartments of photosynthesizing plant cells: β-CA4 in the plasma membrane and in the cytoplasm [12,13], and β-CA2 and β-CA3 in the cytoplasm [12]. Six sources of CA activity were revealed in chloroplasts, including soluble stromal β-CA1 [12,14,15,16], β-CA5 [17,18], and α-CA1 [7,19], as well as membrane-bound α-CA4 [20], and α-CA5 [21] in thylakoids. A CA of the β family was also found in the thylakoid lumen of pea and Arabidopsis chloroplasts [22]. Despite numerous assumptions about the physiological roles of CAs in plant cells [23], none of the hypotheses are considered proven.

Regarding the situation of α-CA2 in plant cells and its presence in different parts of plants, the data are rather contradictory. In 2007, it was shown [12] that the *α-CA2* gene was expressed in Arabidopsis leaves only under reduced CO_2_ levels, but Rudenko et al. [24] have found that this gene was expressed in Arabidopsis leaves under any growth conditions. It was demonstrated that high levels of *OsαCA2* transcripts accumulate in the flag leaves of rice and that α-CA2 was located in the plasma membrane [19]. Wang et al. [25] showed that *LjaCA2* was highly expressed in the root nodules of *Lotus japonicus*, suggesting its potential role in maintaining nodule functioning. Weerasooriya et al. [26] have demonstrated that, in Arabidopsis, the gene of the fluorescent protein fused with the *α-CA2* gene indicated the localization of this CA in the cell wall, while the expression of the correspondent gene was found in both roots and shoots. According to our previous data, one of these chloroplast CAs may be α-CA2 [27,28].

We have shown that α-CA2 mutants exhibited lower proton concentration differences across thylakoid membranes (ΔpH) compared to WT [28]. We have also previously shown that, in α-CA2-KO, the level of plastoquinone (PQ) pool reduction was lower than in WT plants [27]. This correlated with the lower donor side limitation of the PSI (Y(ND)) parameter in these plants as compared to WT. The hydrogen peroxide content in the leaves of mutants lacking α-CA2 after 5 min of high light illumination (500 μmol quanta m^−2^s^−1^) was lower than in the leaves of the WT [27]. Based on the photosynthetic characteristics of α-CA2-KO plants, as well as those of thylakoids isolated from their leaves, we hypothesized [27] that α-CA2 is situated on the stromal side of the thylakoid membrane, and may regulate proton concentration in the thylakoid lumen, thereby indirectly influencing the non-photochemical quenching of chlorophyll *a* fluorescence.

There is evidence that the redox state of the PQ pool induces the signal transduction pathways of the PSII light-harvesting complex’s (LHCII) size regulation [29,30]. Later, it was found that H_2_O_2_ participates in the regulation of LHCII’s size [31]. It is generally accepted that hydrogen peroxide is a very important signaling molecule in plant cells [32,33,34]. Hydrogen peroxide, which is produced within thylakoid membranes as a result of the reduction of superoxide radicals generated in the PSI and the PQ pool by molecules of plastoquinol (PQH_2_)—but not the redox state of the PQ pool per se—was proposed to be the actual carrier of the regulator signals [35]. This fraction of H_2_O_2_ can be conditionally referred to as “membrane” H_2_O_2_. Some data led to the proposition that just this H_2_O_2_ plays a priority role in the above regulation of LHCII’s size [36].

Based on the data in [37,38,39], a scheme of the signal transduction of H_2_O_2_ levels through the retrograde chloroplast–nucleus pathway of the *lhcb1* gene expression’s regulation was suggested [36]. It was assumed that H_2_O_2_ alters the activity of one of the chloroplast proteases by diffusing through the chloroplast envelope [36]. According to the data [40,41], in Arabidopsis chloroplasts there are two proteases that are the most likely participants of this mechanism: ASP—a protease of the envelope, and SPPA1—which is a protease located in non-appressed thylakoid regions. SPPA1 may also be involved in hydrogen peroxide signaling in chloroplasts, since it has been shown that, in Arabidopsis mutants with a T-DNA insertion in the *SPPA* gene, the acclimation processes to high-intensity light was altered compared to the WT [41]. This increase in activity of the chloroplast protease, possibly of ASP, led to the conversion of transcription factor (TF) PTM [42], a homeodomain TF associated with the chloroplast envelope, into a soluble form as a result of proteolysis of this TF, which led to its detachment from the transmembrane domains [38]. This form of PTM, which is released from the chloroplast envelope into the cytoplasm, moves into the nucleus, and activates the expression of the other TF, abscisic acid insensitive 4 (ABI4). ABI4 is able to bind with the promoter sequences of *lhcb* genes in the nucleus, leading it to block their expression [36,43]. Thus, the complete signal transduction pathway from chloroplast to nucleus during regulation of the PSII antenna size can include the following components: hydrogen peroxide, mostly generated in the thylakoid membrane, chloroplast protease, PTM, ABI4, and *lhcb* genes located in the nucleus.

In the present study, we investigated how an *α-CA2* gene knockout affected PSII antenna size, and the possible role of α-CA2 in the above described mechanism of LHCII size regulation.

## 2. Results

### 2.1. Determination of Chlorophyll a and b Content

The content of chlorophyll *a* and *b* in mutant leaves was higher than in the WT, while the chlorophyll *a*/*b* ratio was lower, suggesting a larger PSII antenna size in α-CA2-KO plants (Table 1). The decrease in the Chl *a*/Chl *b* ratio resulted entirely in a higher (31–41%) content of chlorophyll *b* compared to the WT plants. No significant differences in carotenoid content were observed.

### 2.2. Assessment of the Parameters of OJIP Kinetics

The parameter assessment of the OJIP kinetics of chlorophyll a fluorescence of leaves showed that the maximum quantum yield of PSII (Fv/Fm) in the mutants of the two lines studied did not noticeably differ from that parameter in the WT (Figure 1A). That means that the reaction center of PSII α-CA2-KO was not damaged. The parameters characterizing the apparent antenna size, ABS/RC, and energy dissipation into heat, DIo/RC, were slightly higher in line 8–3 and significantly higher in line 9–11 of the mutants compared to the WT (Figure 1B,C).

### 2.3. Estimation of the Amount of Lhcb1, Lhcb2, and D1 Proteins, and Evaluation of the Expression Level of Genes Encoding PSII Antenna Proteins

The contents of the major proteins of LHCII were estimated using Western blot analysis. In the α-CA2-KO plants, the content of Lhcb1 was slightly higher than in the WT (Figure 2A,D), and the content of Lhcb2 was 2–4.5 times higher in both mutant lines than in WT plants (Figure 2B,E). At the same time, in α-CA2-KO plants, the content of the protein D1, which is one of the two main proteins in the PSII core complex, was the same as in the WT (Figure 2C,F).

### 2.4. Measurement of Hydrogen Peroxide Production

The expression level of the *lhcb1* gene was higher in mutants than in WT plants, while the expression levels of the *lhcb2* and *lhcb3* genes were lower in both mutant lines compared to the WT plants (Figure 3).

The measurement of total hydrogen peroxide production in the leaves of WT and mutant plants under an illumination of the same light intensity as during growth revealed that it was twice as low as in the WT (Figure 4).

It has previously been shown that the rate of “membrane” hydrogen peroxide formation increased significantly with increasing light intensity [35]. Therefore, in the present study, high light intensity was used to more clearly reveal the differences in the rate of “membrane” H_2_O_2_ formation between WT plants and α-CA2-KO plants under conditions that resulted in the maximum rate of hydrogen peroxide formation in the thylakoids. To estimate the production of “membrane” H_2_O_2_, cytochrome C, an effective superoxide radical scavenger in the water phase, was added to the thylakoid suspension at a saturating concentration [35] (for more details, see Section 4). While the total rate of H_2_O_2_ production was the same in the WT and α-CA2-KO thylakoids (Figure 5), the rate of “membrane” H_2_O_2_ formation was significantly—about 9 times—lower in the thylakoids of mutant plants than in those of WT plants (Figure 5). It is also worth noting that the effect of the absence of α-CA2 on the level of hydrogen peroxide production in the thylakoid membranes indicates that this CA is indeed situated in the chloroplasts.

### 2.5. Evaluation of the Expression Level of Genes Encoding Proteins Included in Retrograde Signaling

The expression level of the genes encoding the TFs ABI4 and PTM, as well as both chloroplast proteases, ASP and SPPA1 (i.e., the regulation proteins), which were suggested to be the involved in the described retrograde H_2_O_2_ signaling from the chloroplasts to the cell nucleus (see the Introduction), was determined. The expression levels of these genes were significantly (2.5–6.7 times) lower in the mutant plants of both lines than in the WT plants (Figure 6).

## 3. Discussion

A decrease in the Chl *a*/Chl *b* ratio usually corresponds to the increasing size of the light-harvesting antenna [44,45,46], and the fact that the Chl *a*/Chl *b* ratio in α-CA2-KO was lower than in the WT (Table 1) implies a greater antenna size in these mutants than in the WT. Analysis of the OJIP fluorescence kinetics revealed an increase in the apparent PSII antenna size in α-CA2-KO compared to the WT, since the ABS/RC values in the leaves of mutants were higher, especially in the plants of line 9–11 (Figure 1B). Previously, we suggested that the absence of α-CA2 in plants may lead to unregulated proton leakage from the thylakoid lumen into the stroma [28]. The decreased number of protons in the thylakoid lumen of mutants may signal about the lower light intensity than the actual one, prompting α-CA2-KO plants to increase antenna size for more efficient light harvesting.

There are two main types of dissipation of absorbed light energy into heat: regulated and unregulated. The latter type is energy dissipation that occurs during energy migration within pigment–protein complexes of PSII. The OJIP parameter, DIo/RC, reflects the portion of absorbed energy that does not reach the reaction centers to drive photochemical reactions but is instead dissipated as heat [47]. In α-CA2-KO plants, the observed increase in DIo/RC may be associated with an enlarged LHCII size, which could result in a longer migration path for excitation energy and, consequently, greater energy dissipation as heat (Figure 1C).

The contents of the Lhcb1 and Lhcb2 proteins, especially the latter, were higher in mutants than in the WT (Figure 2A,B,D,E). As well, the content of the main PSII core complex protein D1 was not changed in α-CA2-KO plants compared to the WT (Figure 2C,F), indicating the continued stability of the PSII reaction center with an increase in the LHCII size.

Knocking out α-CA2 affected the intensity of the expression of the genes encoding Lhcb1, Lhcb2, and Lhcb3. Specifically, the expression level of the *lhcb1* gene was higher, while the expression levels of lhcb2 and lhcb3 were lower, in mutants compared to the WT (Figure 3). The opposite effect on Lhcb2 content and expression of the *lhcb2* gene was reported in a CA gene knockout study of Arabidopsis plants with a knocked out gene encoding thylakoid α-CA4 [35]. These data are evidence that a change in antenna size as a result of the knockout of genes encoding CA may take place not only at the transcriptional but also at the translational level.

Hydrogen peroxide has previously been shown to be involved in the regulation of expression of genes encoding LHCII proteins [48,49]. Previously, Zhurikova et al. [27] showed that, in high light, the accumulation of hydrogen peroxide in the leaves of α-CA2 mutants was lower than in WT plants. The present study indicates that the observed effects of knocking out α-CA2 on the LHCII size also occur when using low light during plant growth. Under these light conditions, the H_2_O_2_ content in the leaves of mutant plants was considerably lower than in the WT plants (Figure 4).

As previously stated in the Introduction, the level of H_2_O_2_ regulates LHCII’s size [36]. The lower level of H_2_O_2_ accumulation in the leaves of α-CA2-KO plants compared to WT plants satisfactorily explains the increase in the PSII antenna size in mutants. It was therefore proposed that “membrane” H_2_O_2_ may play an important role in the LHCII size regulation [35]. In experiments with isolated thylakoids that were performed using high light illumination, we revealed that, with the total rate of H_2_O_2_ production almost equal in WT and mutant thylakoids, the level of “membrane” H_2_O_2_ formation was significantly lower in the thylakoids of both α-CA2-KO plants lines compared to WT plants (Figure 5). Thus, α-CA2 is able to affect the ratio of the level of H_2_O_2_ formed in reaction of superoxide radicals produced within thylakoid membrane in reaction with PQH_2_ and level of H_2_O_2_ formed in superoxide radicals dismutation reaction. If α-CA2 has the same effect on the ratio of H_2_O_2_ production ways in low light conditions during plant growth, then the higher light-harvesting antenna size in α-CA2-KO plants confirms the assumption in [36] regarding the role of the “membrane” H_2_O_2_ in the regulation of this size.

H_2_O_2_ induces cell signaling cascades leading to the triggering of specific defense responses, including changes in the expression levels of various genes [50]. It has been shown that hydrogen peroxide is able to activate proteases from different organisms [51,52]. We have found that the α-CA2-KO plants exhibited reduced expression levels of genes encoding thylakoid protease, SPPA1, and the ASP protease in chloroplast envelope, as well as genes encoding the TFs ABI4 and PTM (Figure 6). This correlates with the lower hydrogen peroxide production in leaves of mutant compared to WT plants (Figure 4). Since the ABI4 and PTM TFs repress the expression of *lhcb* genes, the decreased gene expression of these TFs in α-CA2-KO plants compared with WT plants is expected to lead to an increase in the size of the PSII antenna of these mutants.

However, as shown earlier, the transcriptional responses induced by the application of exogenous H_2_O_2_ and hormones overlapped substantially, suggesting that both of these signaling pathways may be interconnected [53]. Although CAs catalyze the interconversion of inorganic carbon forms, these enzymes have been implicated as effectors or regulators in salicylic acid (SA), jasmonic acid (JA), and abscisic acid (ABA) signaling cascades [45]. The ability of CAs in the β family to interact with SA, as well as with the key proteins of SA-induced signaling NPR1 and NRB4, has also been shown to be essential for SA perception [54]. In the presence of NPR1 and NRB4, SA inhibits CA activity and triggers the translocation of several CAs from the chloroplast to the cytoplasm, which may constitute a part of the signaling process leading to the nucleus [54]. Data were also obtained on the participation of CA in the synthesis of fatty acids, thus influencing the biosynthesis of JA since fatty acids are the precursors of JA synthesis in chloroplasts. In another study, it was shown that treatment of Arabidopsis plants with JA significantly increased the expression of the genes encoding β-CA1 and β-CA2 [55]. Short-term treatment of pea seedlings with ABA led to a significant increase in CA activity [56]. It is known that ABA, which is often associated with stress conditions (e.g., drought), can reduce the transcription of genes, encoding proteins of LHCII and promote the degradation of LHCII proteins [57,58]. Based on the studies mentioned above, it can be hypothesized that the regulation of PSII antenna size in α-CA2-KO plants may be connected not only to changes in the concentration of H_2_O_2_ in leaves but also with altered hormonal signaling resulting from knocking out α-CA2.

The results of our study indicate that α-CA2 is involved in the regulation of the size of the PSII light-harvesting antenna. Since antenna size is directly related to the efficiency of light capture and adaptation to light, the role of α-CA2 may be of particular importance in the context of the influence of variable light conditions in nature. It has previously been shown that the regulation of PSII antenna size is a universal adaptive mechanism that is triggered not only under high light conditions but also in response to other stresses, such as drought and salinity, even under low light intensity [31,59,60,61]. Increased hydrogen peroxide production and the onset of oxidative stress accompany various abiotic stress factors, such as drought, salinity, and changes in CO_2_ concentrations [62,63,64]. Thus, it can be assumed that the observed effect of α-CA2 functioning on the regulation of the size of the PSII light-harvesting antenna may play an important role in plant adaptation not only to high light conditions but also to various abiotic stress factors.

## 4. Materials and Methods

### 4.1. Plant Material and Growth Conditions

*A. thaliana* plants of the Columbia ecotype (WT) and plants with a knocked out *At2g28210* gene encoding α-CA2 (α-CA2-KO) (the homozygous line “9–11”, obtained from the SALK_120400 line, and the homozygous line “8–3”, derived from the SALK_080341C line) were used. A schematic representation of the T-DNA insertions in the *At2g28210* gene encoding α-CA2 is shown in Appendix A. SALK_120400 is the position of the insertion in the homozygous mutant plant “9–11” line (Appendix A), and SALK_080341C is the position of the insertion in the homozygous mutant plant of the “8–3” line (Appendix A).The results of the 1.5% agarose gel electrophoresis of PCR products with primers to the *α-CA2* gene were performed according to [27], and cDNA obtained after the reverse transcription of RNA isolated from the leaves of WT plants and the 8–3 and 9–11 mutant lines showed the absence of *α-CA2* gene transcripts in both mutant plant lines (Appendix A). The plant seeds were kindly provided by Prof. J.V. Moroney (Louisiana State University, Baton Rouge, LA, USA). After the formation of the first four true leaves, at 14–21 days of age, the plants were transplanted into 150 mL pots with soil. The plants were grown in a climate chamber at a constant temperature of 21–22 °C, an illumination of 50–70 μmol quanta m^−2^s^−1^and a CO_2_ concentration of 450 ppm. For the experiments, 45–50-day-old plants were used.

### 4.2. Measurement of Chlorophyll a Fluorescence

To measure the OJIP chlorophyll a fluorescence transient in leaves, a Handy Pea fluorometer (Hansatech, Pentney, UK) was used to record the fluorescence during a 1 s flash of red light of 3000 μmol quanta m^−2^s^−1^. The measurements were conducted on plants that had been dark-adapted for 2 h. From these measurements, the parameter characterizing the apparent antenna size, ABS/RC, and energy dissipation into heat, DIo/RC were calculated according to [65].

### 4.3. Determination of Chlorophyll and Carotenoid Content

The chlorophyll content was determined in 96% ethanol extracts [66].

### 4.4. Western Blot Analysis

Immunoblotting was performed as described in the protocol of BioRad laboratories. The separation of proteins of the thylakoid membrane, isolated from leaves of 5 to 6-week-old WT or α-CA2-KO Arabidopsis plants, was carried out by electrophoresis in a 16% polyacrylamide gel in a Mini-PROTEAN Cell (BioRad, Hercules, CA, USA) under denaturing conditions. PageRuler™ Prestained Protein Ladder (10–180 kDa) (Thermo Fisher Scientific, Waltham, MA, USA) was used as a protein molecular mass marker. After electrophoresis, proteins were transferred onto a PVDF membrane (BioRad, Hercules, CA, USA) using the Mini Trans-Blot Cell (BioRad, Hercules, CA, USA) wet blotting system. Primary polyclonal rabbit antibodies against Lhcb1, Lhcb2, and D1 proteins (Agrisera, Vännäs, Sweden) were used. Goat anti-rabbit IgG, AP conjugated (BioRad, Hercules, CA, USA), was used as the secondary antibody. An alkaline phosphatase conjugate substrate kit (BioRad, Hercules, CA, USA) was used for visualization. Western blot analysis results were obtained from six independent experiments. The membranes were scanned for further analysis. Quantification of bands on the blots was performed by the Image J 1.54g software OriginPro, 2021 package.

### 4.5. Quantitative Reverse Transcription PCR

Total RNA was extracted from frozen Arabidopsis leaves using the kit R-plants (Biolabmix, Novosibirsk, Russia) with DNase treatment to eliminate any genomic DNA contamination. Complementary DNA synthesis was performed using the reverse transcription kit OT-1 (Sintol, Moscow, Russia) with oligo (dT) as a primer. Quantitative reverse transcription polymerase chain reaction (qRT-PCR) was performed with qPCRmix-HS SYBR (Evrogen, Moscow, Russia). The sequences of primer used in this study are presented in Table 2. qRT-PCR data were normalized against the housekeeping *Ubiquitin* gene. PCR reactions were run in a LightCycler 96 Instrument from Roche Diagnostics GmbH (Rotkreuz, Switzerland).

### 4.6. Measurement of Hydrogen Peroxide Content in Leaves

To measure the H_2_O_2_ content, plant leaves were taken from the growth chamber after 3 h of illumination, weighed, and frozen in liquid nitrogen. Then they were ground in 0.4 mL of 2 M trichloroacetic acid (TCA) and the homogenate was transferred to test tubes with 3 mL of 0.05 M potassium phosphate buffer. Activated carbon was added to the test tubes to remove the chlorophylls, pheophytin, and carotenoids. After 1 h of incubation at 4 °C, the homogenate was centrifuged at 10,000× *g* for 10 min. The supernatant was titrated with 2 M KOH to pH 7.0. The hydrogen peroxide content was measured in 50 μL of the supernatant using a Lum-101 device (Moscow, Russia) by the luminescence of a mixture of 5 μM horseradish peroxidase and 50 mM luminol. When constructing the calibration curve, 50 µL of H_2_O_2_ of a known concentration was used instead of the supernatant.

### 4.7. Measurement of the Light-Induced Changes of Oxygen Concentration in a Suspension of Isolated Thylakoids

Thylakoids were isolated from Arabidopsis leaves according to [67]. The rate of oxygen evolution/uptake was measured in a thermostatic glass cell with a volume of 0.3 mL at 21 °C using a Clark-type pO_2_ electrode connected to the computer via ADC. The reaction mixture in the pO_2_ electrode vessel was illuminated with a red LED with a light intensity of 450 μmol quanta m^−2^s^−1^. The basic reaction mixture contained 0.1 M sucrose, 20 mM NaCl, 5 mM MgCl_2_, 50 mM HEPES-KOH (pH 7.6), and thylakoids corresponding to 10 μg of Chl (mL)^−1^. The total O_2_ uptake was measured in the presence of the uncoupler gramicidin D (GrD) (1 μM). The production of “membrane” H_2_O_2_ was determined by measuring the oxygen evolution rate in the thylakoid suspension in the presence of cytochrome C at a 60 μM concentration; as shown earlier, at this concentration, cytochrome *C* scavenges all superoxide radicals, preventing the formation of hydrogen peroxide in the aqueous phase [35]. In all experiments, 1 μM GrD, 60 μM cytochrome C, and 100 units/mL of catalase were used to assess the “membrane” hydrogen peroxide. The difference between the oxygen evolution rate in the presence and absence of catalase was used to estimate the amount of “membrane” H_2_O_2_ formed within the thylakoid.

### 4.8. Statistical Analysis

Statistical analysis was performed using OriginPro, 2021 software. Data are presented as mean values with standard errors of the mean. Statistical significance was assessed using analysis of variance (ANOVA) with a paired comparison plot and the Holm–Bonferroni test.

## 5. Conclusions

The absence of α-CA2 led to an increase in the size of the PSII antenna, as evidenced by an increase in the Chl *a*/Chl *b* ratio and in the ABS/RC ratio, as well as by a higher content of the light-harvesting proteins Lhcb1 and Lhcb2. Knocking out α-CA2 also affected the expression of genes encoding the proteins of the LHCII complex: Lhcb1, Lhcb2, and Lhcb3. It was found that the content of hydrogen peroxide in the leaves of plants with the knocked out α-CA2 encoding gene was lower than in the WT plants, and this presumably led to a change in the size of LHCII through a retrograde H_2_O_2_ signaling pathways. It was shown that α-CA2 affects the share of H_2_O_2_ generated within the thylakoid membrane. The changes revealed in the functioning and structure of the photosynthetic electron transport chain in the Arabidopsis plants with the α-CA2 knockout indicate that this CA is located in the chloroplasts of these plants.

## Figures and Tables

**Figure 1 plants-14-01529-f001:**
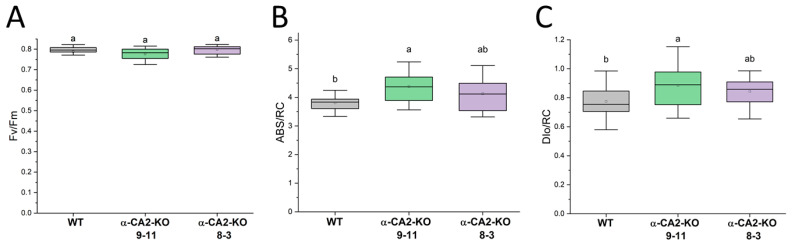
Parameters derived from the OJIP kinetics for *A. thaliana* leaves: Fv/Fm (**A**), ABS/RC (**B**), and DI_0_/RC (**C**) in the WT (grey boxes) and α-CA2-KO plants (line 9–11—green box, line 8–3—purple box). Fv/Fm—the PSII maximal quantum yield, DIo/RC—the quantum yield of energy dissipation, and ABS/RC—characterizing the apparent antenna size. Data are presented as box charts, where boxes are ranges of values that accounted for 50% of all measurements (between 25 and 75% quartiles); the mean values are indicated by dots inside the boxes; the horizontal line inside the box is the median. Different letters indicate significant differences at *p* < 0.05, *n* ≥ 20.

**Figure 2 plants-14-01529-f002:**
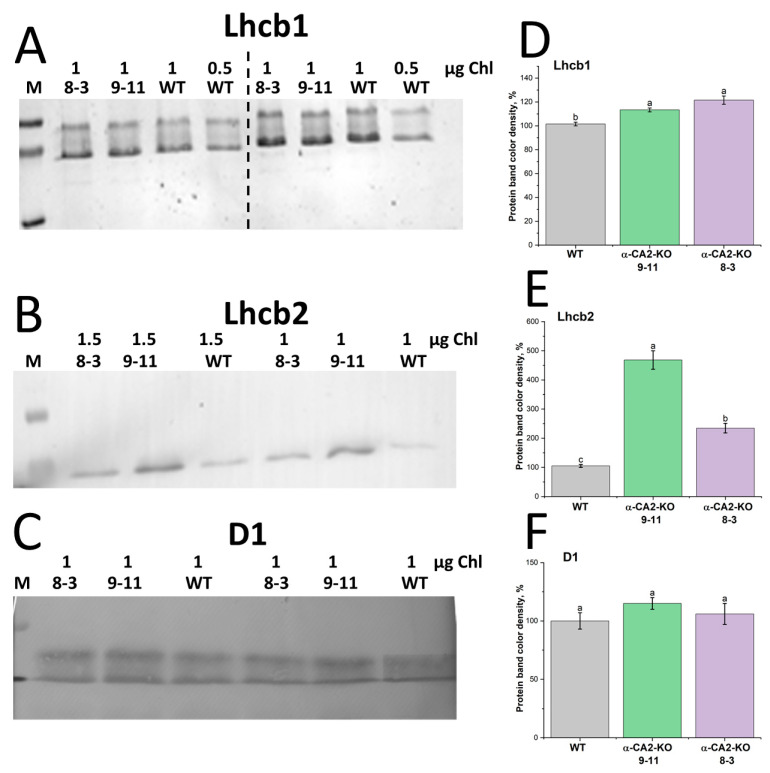
The result of immunoblotting after the denaturing electrophoresis of the thylakoid membranes isolated from leaves of WT (grey bars) and α-CA2-KO plants (line 9–11—green bars, line 8–3—purple bars), which were grown at a light intensity of 50 μmol quanta m^−2^s^−1^ with antibodies against Lhcb1 (**A**), Lhcb2 (**B**), and the major protein of PSII, D1 (**C**). Numbers on the top of the photos of the PVDF membranes indicate the chlorophyll content (μg) in the samples. The data for the mutant lines are given as a percentage of those in the WT. In panel (**A**), thylakoid membranes from plants grown in two independent experiments are shown on the left and right sides of the dashed line. (**B**,**C**)—preparations of thylakoid membranes isolated from plants of one cultivation. The results of densitometric analysis of these proteins are presented as percentages in panels (**D**), (**E**) and (**F**), respectively. The experiments were performed six times. Data are shown as the mean ± SE. Different letters indicate significant differences at *p* < 0.05.

**Figure 3 plants-14-01529-f003:**
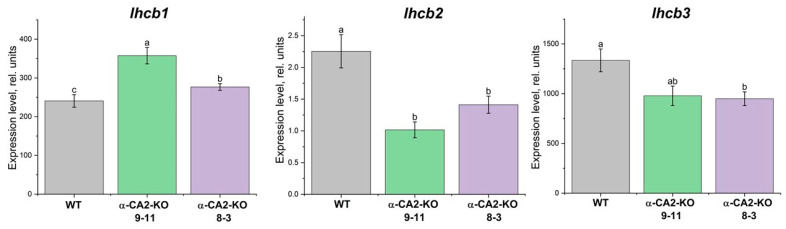
Expression levels of genes encoding PSII light-harvesting complex proteins in WT plants (grey bars) and α-CA2-KO plants (line 9–11—green bars, line 8–3—purple bars) grown at 50 μmol quanta m^−2^s^−1^. Data are shown as the mean ± SE. Different letters indicate significant differences at *p* < 0.05.

**Figure 4 plants-14-01529-f004:**
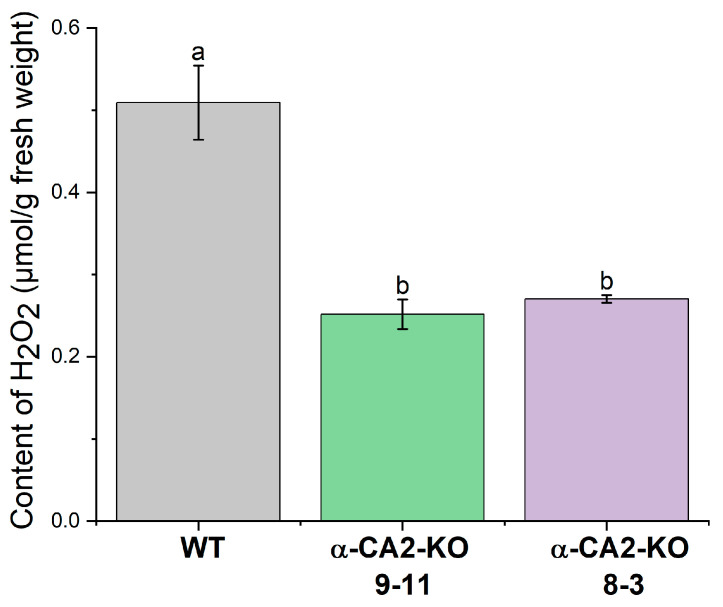
The content of hydrogen peroxide in leaves of WT plants (grey bars) and α-CA2-KO plants (line 9–11—green bars, line 8–3—purple bars) after three hours of illumination at 50 μmol quanta m^−2^s^−1^. Data are shown as mean ± the SE. Different letters indicate significant differences at *p* < 0.05.

**Figure 5 plants-14-01529-f005:**
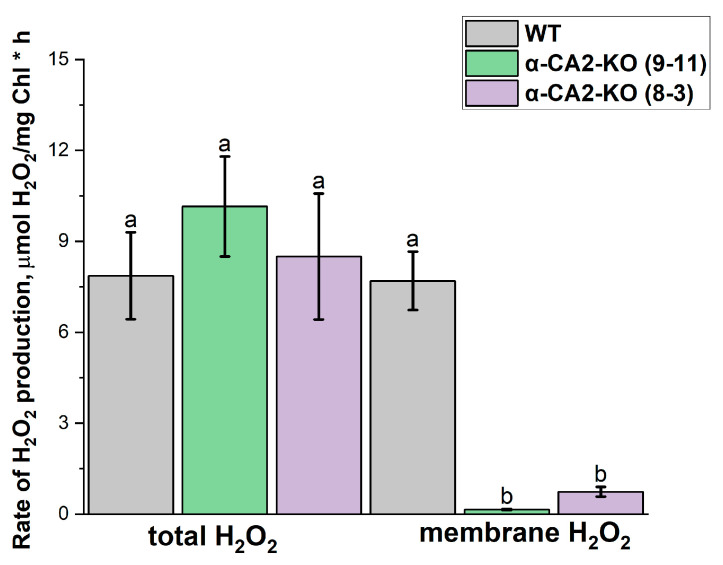
The rate of H_2_O_2_ formation without (total H_2_O_2_) and in the presence of 60 μM of cytochrome C (membrane H_2_O_2_) under high light illumination (500 μmol quanta m^−2^s^−1^) in the thylakoids isolated from WT plants (grey bars) and α-CA2-KO plants (line 9–11—green bars, line 8–3—purple bars) grown at 50 μmol quanta m^−2^s^−1^. Data are shown as the mean ± SE. Different letters indicate significant differences at *p* < 0.05.

**Figure 6 plants-14-01529-f006:**
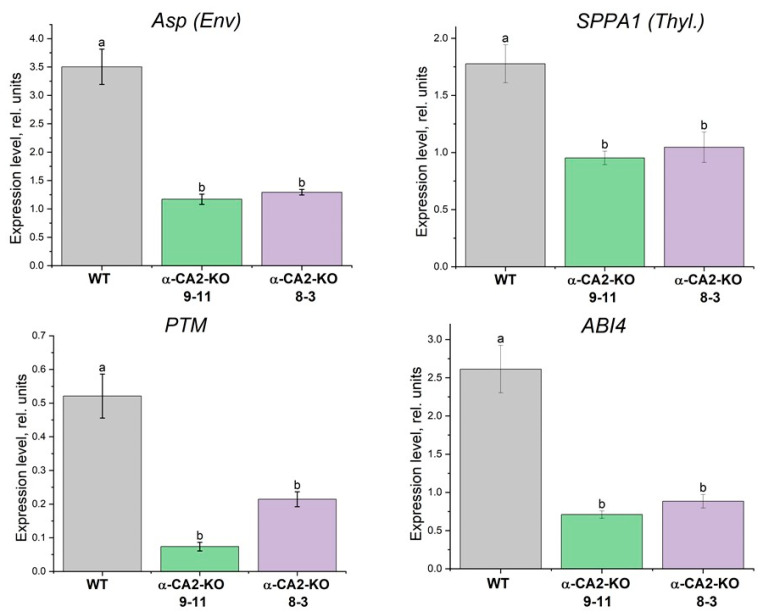
Expression levels of the genes encoding ABI4, PTM, ASP, and SPPA1 in the WT plants (grey bars) and α-CA2-KO plants (line 9–11—green bars, line 8–3—purple bars) grown at 50 μmol quanta m^−2^s^−1^. Data are shown as the mean ± SE. Different letters indicate significant differences at *p* < 0.05.

**Table 1 plants-14-01529-t001:** Content of Chl *a*, Chl *b*, and carotenoids, and the Chl *a*/Chl *b* ratio in the leaves of WT plants and α-CA2-KO mutants grown under an illumination of 70 μmol quanta m^−2^s^−1^. Data are shown as the mean ± SE.

Plants	Pigment Content (mg/g Fresh Weight)			
	Chl *a*	Chl *b*	Chl *a*/Chl *b*	Carotenoids
WT	0.71 ± 0.06	0.29 ± 0.02	2.45 ± 0.06	0.15 ± 0.02
α-CA2-KO (9–11)	0.87 ± 0.09	0.41 ± 0.04 *	2.12± 0.01	0.17 ± 0.02
α-CA2-KO (8–3)	0.89 ± 0.07	0.38 ± 0.02 *	2.34 ± 0.01	0.18 ± 0.02

Significant differences are indicated by *, *p* ≤ 0.05, *n* ≥ 3.

**Table 2 plants-14-01529-t002:** Primers used for qRT-PCR. F is the forward and R is the reverse primer.

Genes	Nucleotide Sequences of Primers	
*At1g73990* (*Arabidopsis Serin Protease* (*SPPA*) gene)	F	TCATTCTCGTGGTCTAATAGATGCTGTC
	R	CGT CGA GCA GTC CTT TTA ATG TTC TG
*At2g32480* (*Arabidopsis Serin Protease* (*ASP*) gene)	F	TGTGGGAAGGGAGTTTATGGGG
	R	GCTGCGAATTGGTAAAGCCC
*At5g35210* (Arabidopsis *PTM* gene)	F	TGA AAAGGGTCTGAGATATTCATATAA GAGATCA
	R	GAGCACTCTGAGTCCAAGCAT
*At2g40220* (Arabidopsis *ABI4* gene)	F	GTTGGAGATGGATCTTCGACCATTT
	R	TTG ACC GAC CTT AGG GAT GCT
*At1g29930* (Arabidopsis *Lhcb1* gene)	F	AGCTCAAGAACGGAAGATTGG
	R	GCCAAATGGTCAGCAAGGTT
*At2g05070* (Arabidopsis *Lhcb2* gene)	F	GTCCATACCAGATGCTTTGGGGAG
	R	CTCACACTCTCTCTTCAATCCTTTCCTTTCAT
*At5g25760* (Arabidopsis *Ubiquitin* gene)	F	TGCTTGGAGTCCTGCTTGGA
	R	TGTGCCATTGAATTGAACCCTCT

## Data Availability

The datasets generated during and/or analyzed during the current study are available from the corresponding author upon reasonable request.

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
