# Peer review of "Effect of the Absence of α Carbonic Anhydrase 2 on the PSII Light-Harvesting Complex Size in Arabidopsis thaliana"

_plants, 2025, doi:10.3390/plants14101529_

Round 1
Reviewer 1 Report
Comments and Suggestions for Authors
I have carefully reviewed this manuscript investigating the role of α-carbonic anhydrase 2 (α-CA2) in regulating photosystem II (PSII) antenna size in Arabidopsis thaliana. The authors present evidence that knocking out the α-CA2 gene leads to an enlarged PSII antenna size, decreased hydrogen peroxide accumulation, and altered expression of genes involved in retrograde signaling pathways.
The study addresses an important aspect of photosynthetic regulation and provides valuable insights into the potential role of carbonic anhydrases in chloroplast function beyond CO2 fixation. The experimental approach is generally sound, combining physiological, biochemical, and molecular techniques to characterize the α-CA2 knockout phenotype.
However, several aspects of the manuscript require significant improvement:
(1)The mechanistic link between α-CA2 function and H2O2 generation remains speculative. Additional experiments exploring how α-CA2 specifically affects the formation of "membrane" H2O2 would strengthen the manuscript's conclusions.
(2)The authors use two different α-CA2-KO lines but do not consistently distinguish between them throughout the results and discussion. Greater clarity regarding line-specific differences (or lack thereof) would enhance the reliability of the findings.
(3)The discussion would benefit from more thorough consideration of alternative mechanisms by which α-CA2 might influence antenna size, beyond the H2O2 signaling pathway proposed.
Statistical analysis should be more rigorously applied across all experiments, particularly for the protein quantification data from Western blots.
The authors should provide a clearer explanation of how the observed changes in gene expression (increased lhcb1 but decreased lhcb2 and lhcb3) collectively result in an enlarged PSII antenna.
The broader implications of these findings for understanding plant responses to changing environmental conditions (particularly light and CO2 levels) should be more thoroughly discussed.
In conclusion, this manuscript presents novel and potentially significant findings regarding the role of α-CA2 in chloroplast function and photosynthetic regulation. With substantial revisions to address the mechanistic aspects and strengthen the data analysis, this work would make a valuable contribution to the field.
Author Response
To reviewer 1
We are very grateful to the reviewer for his constructive comments on our work. We tried to explain more clearly a number of points that the reviewer pointed out. We have also introduced the changes in the text to outline some points in more detail.
Replies to Reviewer’s comments
Comments 1
The mechanistic link between α-CA2 function and H2O2 generation remains speculative. Additional experiments exploring how α-CA2 specifically affects the formation of "membrane" H2O2 would strengthen the manuscript's conclusions.
Reply
At first, we may note that there are literature data, which demonstrate a relationship between CA activity and the production of reactive oxygen species (ROS). For example, Arabidopsis mutants with knockout genes encoding mitochondrial gamma-CA1x gamma-CA2 showed the increased ROS accumulation in mitochondria (Cordoba et al., 2016). Previously, we got the data implying that hydrogen peroxide can affect the level of expression of genes encoding Lhcb proteins, regulating the size of the light-harvesting complex PSII under changing external conditions (Mubarakshina and Ivanov, 2010; Borisova-Mubarakshina et al., 2015; Borisova-Mubarakshina et al., 2019). This was one of main reasons to evaluate the effect of α-CA2 absence on Hâ‚‚Oâ‚‚ formation in the course of studying this effect on the PSII light-harvesting complex size. Indeed, in the present study we have found that the total content of hydrogen peroxide in the leaves of the mutant plants was significantly, twice, lower than in the leaves of WT plants (Fig. 4). Thus, the lower total H2O2 content in α-CA2-KO plants correlates with the larger size of the LHCII, which is consistent with previously obtained data.
As to the formation of the "membrane" H2O2. In our previous study (Borisova-Mubarakshina et al., 2020) we put proposition basing on some observations, that just H2O2 molecules, which are formed in the membrane in the reaction of plastohydroquinone with superoxide radicals generated in photosynthetic electron transport chain (PETC) can be real substance which transfer signal about PETC state to genes encoding Lhcb proteins. Having obtained the result that the formation of “membrane” H2O2 in the thylakoids from α-CA2-KO plants is noticeably lower than in the thylakoids from WT plants (Fig. 5), we certainly will further investigate the mechanism of this phenomenon in more detail. This will help us to better understand the mechanism of α-CA2 operation in the chloroplasts. At this time, we may only speculate why the absence of α-CA2 lead to lower production of “membrane” H2O2, taking into account our previous results.
In the previous study on α-CA2 function in chloroplast we have shown that α-CA2-KO mutants exhibited lower proton concentration difference across thylakoid membranes (ΔpH) than WT plants (Nadeeva et al., 2023). On the basis of these and other data there, we supposed that α-CA2 is located on the stromal side of a proton channel or antiporter in the thylakoid membrane, and is involved in the regulation of proton transport across membrane, i.e., inside the membrane. The “membrane” peroxide is formed in the reaction of plastohydroquinone with superoxide radical. The transfer of protons to the latter can lead to formation of perhydroxyl radical, HO2•, which has higher than superoxide radical activity in the reaction of hydrogen atom detachment from organic molecules and has much higher permeability coefficient in lipid membranes. This may be an important step in the promotion of formation of hydrogen peroxide in the membrane. Thus, basing on the assumption made in (Nadeeva et al., 2023) we hypothesize that α-CA2 may give up protons, which are necessary for the formation of hydrogen peroxide. If this CA is absent, this reaction is slowed down, which is reflected in a lower rate of “membrane” H2O2 formation.
Comments 2
The authors use two different α-CA2-KO lines but do not consistently distinguish between them throughout the results and discussion. Greater clarity regarding line-specific differences (or lack thereof) would enhance the reliability of the findings.
Reply
In this paper, two independent α-CA2-KO mutant lines (SALK_120400 – line 9-11 and SALK_080341C – line 8-3) were used and the results obtained with them were analyzed in parallel. In the supplementary materials, we have added the information about the positions of the knockout insersions in α-CA2 genes in both mutant lines and the results of electrophoresis, which show that cDNA correspondent to α-CA2 encoding gene is absent in these mutant plant lines.
In some experiments, such as measuring OJIP fluorescence parameters (Fig. 1B, 1C), quantitative differences between the lines were revealed: line 9-11 showed a more pronounced increase in ABS/RC and DIo/RC compared to 8-3. However, in most other experiments, including chlorophyll content and Chl a/b ratio (Table 1), Lhcb1 and Lhcb2 protein levels (Fig. 2), lhcb1, lhcb2, lhcb3, ABI4, PTM, SPPA1, ASP genes expression (Figs. 3 and 6), as well as total and "membrane" Hâ‚‚Oâ‚‚ levels (Figs. 4 and 5), both lines showed similar direction and magnitude of effects compared to WT, indicating high reproducibility of the phenotype. We agree that this should be more clearly stated. In a revised version of the manuscript, we clarify what differences and similarities were observed between the mutant lines and summarize this in the Discussion section.
Comments 3
The discussion would benefit from more thorough consideration of alternative mechanisms by which α-CA2 might influence antenna size, beyond the H2O2 signaling pathway proposed.
Reply
It is a generally accepted that hydrogen peroxide is a very important signaling molecule in plant cell. As it is noted in the reply to Comment 1, a hydrogen peroxide-dependent retrograde signaling pathway plays possibly a predominant role in regulating PSII antenna size (Mubarakshina and Ivanov, 2010; Borisova-Mubarakshina et al., 2015; Borisova-Mubarakshina et al., 2019). Thus, we pay main attention to this mechanism underlying the relationship between α-CA2 absence and antenna size. The review of other ways of the antenna size regulation would be mostly speculative. However, in our future investigation and papers we will use your advice to considerate the alternative mechanisms.
Comments 4
Statistical analysis should be more rigorously applied across all experiments, particularly for the protein quantification data from Western blots.
Reply
We have completed the missing statistical processing of the data. Data Western blots (Fig. 2 D–F) now are presented as means ± SE, and statistical significance was determined using ANOVA followed by the Holm-Bonferroni test.
Comments 5
The authors should provide a clearer explanation of how the observed changes in gene expression (increased lhcb1 but decreased lhcb2 and lhcb3) collectively result in an enlarged PSII antenna.
Reply
It is known that the changes, which take place at the level of gene expression do not always coincide with changes at the level of protein (Gry et al., 2009 (https://doi.org/10.1186/1471-2164-10-365); Liu et al., 2016 (https://doi.org/10.1016/j.cell.2016.03.014). The process of translation can affect the level of protein. We have found that the knockout of the α-CA2 gene results in an increase in the content of two major PSII antenna proteins, Lhcb1 and Lhcb2 (Fig. 2), in spite of the expression level of the gene encoding Lhcb1 increased and the expression level of the gene encoding Lhcb2 decreased (Fig. 3).
Comments 6
The broader implications of these findings for understanding plant responses to changing environmental conditions (particularly light and CO2 levels) should be more thoroughly discussed.
Reply
Thanks for the comment, that spurred us to update the Discussion. The results of our study indicate that α-CA2 is involved in the regulation of the size of the PSII light-harvesting antenna. Since antenna size is directly related to the efficiency of light capture, and adaptation to light, the role of α-CA2 can be of particular importance in the context of influence of variable light conditions in nature. It has previously been shown that the regulation of PSII antenna size is a universal adaptive mechanism that is triggered not only under high light conditions but also in response to other stresses, such as drought and salinity, even under low light intensity (Borisova-Mubarakshina et al., 2020). The increased hydrogen peroxide production and the onset of oxidative stress accompany the various abiotic stress factors. It is worth noting that changes in CO2 in the environment are likely to affect the functioning of carbonic anhydrase, which catalyzes the reaction of reversible hydration of CO2. Based on this, it can be assumed that the observed effect of α-CA2 on hydrogen peroxide formation may play a certain role in plant adaptation not only to high light but changed concentration CO2 conditions in environment.
Reviewer 2 Report
Comments and Suggestions for Authors
The authors discuss the carbonic anhydrases mutant plants to discover that the changes in the size of the PSII light-harvesting complex in the absence of α-CA2 correlate with a decreased accumulation of Hâ‚‚Oâ‚‚ in the leaves of the mutants. However, significant revisions are necessary to strengthen the manuscript. Below are my detailed comments and suggestions:
- Confirmation of the CA2-KO mutant should be provided, such as through RNA level detection. Additionally, morphological differences between the mutant and normal plants, differences in leaf symptoms, and differences in photosynthetic efficiency should be addressed.
- Regarding the selection of Lhcb1 and Lhcb2, it is important to note that there are over 22 Lhca/b genes in Arabidopsis thaliana. Why were these two specifically chosen?
- Adding subheadings to the results section would enhance understanding.
- The content would be more appropriately written as a briefing.
- Table 1 shows that the difference between Chla WT and the mutant is 0.16 - 0.18, while the difference between Chlb WT and the mutant is 0.12 - 0.09. Why is the difference in Chlb significant, but not in Chla?
- The Western Blot (WB) results in Figure 2 are missing an internal control, making it impossible to rule out differences in the loading amount.
Author Response
To reviewer 2
Dear Reviewer, we are very appreciative of a detailed review of our manuscript. The reviewer's comments are very useful for a better presentation of our data. We took advantage of all the reviewer's advices. Below, we listed the replies to every reviewer's comment.
Replies to Reviewer’s comments
Comments 1
Confirmation of the CA2-KO mutant should be provided, such as through RNA level detection. Additionally, morphological differences between the mutant and normal plants, differences in leaf symptoms, and differences in photosynthetic efficiency should be addressed.
Reply
The effectiveness of α-CA2 knockout in mutant lines was confirmed in previously published papers (Zhurikova et al. 2016, Nadeeva et al. 2023) and in this work using several approaches. The expression of the α-CA2 gene was analyzed using quantitative RT-PCR. It was shown that in both mutant lines (8-3 and 9-11), the transcripts of α-CA2 were absent in contrast to the wild type plants (WT). In the supplementary materials, we have added the information about the positions of the knockout insertions in α-CA2 genes in both mutant lines and the results of electrophoresis, which show that cDNA correspondent to α-CA2 encoding gene is absent in these mutant plant lines.
Visual assessment of the plants did not reveal any noticeable differences in morphology between the mutants and WT under normal growing conditions. We have previously shown (Zhurikova et al. 2016, Nadeeva et al. 2023) that α-CA2-KO plants did not exhibit deviations in growth and development under controlled light, which is also consistent with the present work.
Changes in photosynthetic activity were assessed by the level of chlorophyll a fluorescence using Dual-PAM and OJIP kinetics. It was previously shown that α-CA2 mutants exhibited an increase in the rate of electron transport through PSII and PSI, lower proton concentration difference across thylakoid membranes (ΔpH) and lower the level of plastoquinone (PQ) pool reduction than in WT plants (Nadeeva et al. 2023). In the present work, Fig. 1 shows that the mutants have increased light absorption per reaction center (ABS/RC) and increased the proportion of dissipated energy (DIo/RC). These parameters differ from WT and confirm functional changes in the photosynthetic apparatus. Thus, the efficiency of α-CA2 knockout is confirmed at the molecular level, as well as by functional characteristics, despite the lack of a pronounced morphological phenotype.
Comments 2
Regarding the selection of Lhcb1 and Lhcb2, it is important to note that there are over 22 Lhca/b genes in Arabidopsis thaliana. Why were these two specifically chosen?
Reply
The Lhc a/b gene family in Arabidopsis thaliana is really large. In this study, the main focus was on the analysis of the light-harvesting antenna of photosystem II (PSII), which is made up of Lhcb proteins. Of all the Lhcb proteins, we chose Lhcb1 and Lhcb2 because they are the main structural components of the trimeric LHCII complexes, which compose the majority of the PSII antenna. In addition, the genes of Lhcb1 and Lhcb2 are the most abundantly expressed genes, and variation in their expression play a key role in regulating antenna size. Many studies (Anderson, 1986; Lindahl et al., 1995; Bailey et al., 2001; Żelisko et al., 2005; Ballottari et al., 2007; Wagner et al., 2011) have shown that these proteins demonstrated the greatest differences in level expression of their genes and protein content in processes of light adaptation and could be sensitive markers of changes in the light-harvesting antenna of PSII.
Comments 3, 4
Adding subheadings to the results section would enhance understanding. The content would be more appropriately written as a briefing.
Reply
We have added subheadings to the results section.
Determination of chlorophylls a and b contents
The assessment of the parameters of OJIP kinetics
Estimation of the amount of lhcb1, Lhcb2 and D1 proteins and evaluation of the expression level of genes encoding PSII antenna proteins
The measurement of hydrogen peroxide production
Evaluation of the expression level of genes encoding proteins included in retrograde signalling
Comments 5
Table 1 shows that the difference between Chla WT and the mutant is 0.16 – 0.18, while the difference between Chlb WT and the mutant is 0.12 – 0.09. Why is the difference in Chlb significant, but not in Chla?
Reply
Table 1 indeed shows that the absolute differences in chlorophyll a (Chl a) content between WT and mutants are of 0.16–0.18 mg/g, while the differences in chlorophyll b (Chl b) were of 0.09–0.12 mg/g. However, statistical significance was determined. In the case of Chl a, there was a large intra-group variability, especially in the control group (WT), which reduced the reliability of differences in statistical analysis. In contrast, Chl b values in both mutant lines showed a stable and more homogeneous decrease, which made it possible to identify a statistically significant difference even with a smaller absolute change.
Comments 6
The Western Blot (WB) results in Figure 2 are missing an internal control, making it impossible to rule out differences in the loading amount.
Reply
Thank you for your comment. We fully share your concern about the need for proper loading control when performing Western blot analysis.
In this work we used dilutions according to chlorophyll content as a loading control, which is a generally accepted approach for the analysis of thylakoid proteins. The data are shown in Figure 2A. Figure 2 shows typical Western blot results. It should be noted that at least six independent biological replicates were performed for each studied protein, the results of which confirm the observed patterns.
Round 2
Reviewer 1 Report
Comments and Suggestions for Authors
I have thoroughly reviewed the manuscript "Effect of the Absence of α Carbonic Anhydrase 2 on the PSII Light-Harvesting Complex Size in Arabidopsis thaliana" along with the authors' responses to the previous reviewer comments. The authors have made significant improvements to the manuscript, addressing the concerns raised with additional explanations, clarifications, and statistical analysis. The study provides insights into the role of α-CA2 in regulating the size of the PSII antenna through hydrogen peroxide signaling pathways in Arabidopsis thaliana.
The authors have clarified how the two independent mutant lines were analyzed throughout the study and added supplementary information about insertion positions. They effectively address line-specific differences where observed while noting the overall consistency in phenotype across most measurements, which strengthens the reliability of their findings. The statistical treatment of the data has been improved, particularly for the Western blot data.
Regarding the apparent discrepancy between gene expression and protein levels, the authors have provided a satisfactory explanation, noting that translation processes can affect final protein content. This clarification appropriately addresses the complex relationship between transcription and translation, which is an important consideration in interpreting their results.
However, I strongly encourage the authors to expand their discussion of alternative mechanisms by which α-CA2 might influence antenna size beyond the Hâ‚‚Oâ‚‚ signaling pathway. Even if speculative, considering alternative hypotheses would enrich the paper and potentially inspire new research directions.
Author Response
To Reviewer 1
We are very grateful to the reviewer for his work in reviewing our manuscript.
We have added to the Discussion section (lines 254-274) a text devoted to the hypothesis of an alternative mechanism for regulating the size of the PSII antenna.
Reviewer 2 Report
Comments and Suggestions for Authors
That is OK.
Author Response
To Reviewer 2
We are very grateful to the reviewer for his work in reviewing our manuscript.